# Study on the Radioactivity Levels of Metal Tailings in the Lhasa Area of Tibet

**DOI:** 10.3390/ijerph20054525

**Published:** 2023-03-03

**Authors:** Rengui Weng, Feng Tian, Guohong Chen, Shuo Dong, Junjiang Bai

**Affiliations:** College of Ecological Environment and Urban Construction, Fujian University of Technology, Fuzhou 350118, China

**Keywords:** concentration of active, absorbed dose rate, metallic tailings, radioactive level

## Abstract

The main purpose of this study was to determine the natural radioactivity level of raw radionuclides in the metal tailings of a mine in Lhasa, Tibet, and to conduct sampling and detection in 17 typical metal tailing mines in Lhasa, Tibet. The specific activity concentrations of ^226^Ra, ^232^Th, and ^40^K in the samples were calculated. The total αβχγ radiation, radon concentration, and outdoor absorbed dose rate in the air 1.0 m above the ground were measured. The γ radiation levels affecting miners and their surrounding residents were assessed. The results show that the radiation dose ranges from 0.08 μSv/h to 0.26 μSv/h, and the radon concentration ranges from 10.8 Bq/m^3^ to 29.6 Bq/m^3^, which does not exceed the national radiation-related standards, and the environmental hazard risk is low. The specific activity concentration of ^226^Ra ranged from 8.91 Bq/kg to 94.61 Bq/kg, the specific activity concentration of ^232^Th ranged from 2.90 Bq/kg to 89.62 Bq/kg, and the specific activity concentration of ^40^K was less than MDA to 762.89 Bq/kg. The average absorbed dose rate (DO) of the 17 mining areas was 39.82 nGy/h, the average annual effective dose rate (EO) was 0.057 mSv/y. The average external risk index of the 17 mining areas was 0.24, the average internal risk index was 0.34, and the average γ index was 0.31, all of which were less than the maximum permissible limit. This means that the metal tailings from all 17 mining areas were within the limit for γ radiation and, therefore, can be used in bulk as major building materials without posing a significant radiation threat to the residents of the study area.

## 1. Introduction

China is the third largest mining country in the world, and mining and utilization have caused many environmental problems [1,2,3]. The environmental regulations of the People’s Republic of China on the Prevention and Control of Radioactive Pollution give a clear definition of the associated radioactive mines, but the definition standard of associated radioactive mines has been perplexing for the departments of ecological and environmental protection, the nuclear industry, and the exploration and development of various resources, and it is still under the process of long-term investigation and research. According to the survey, the output of industrial solid waste in China exceeded 3 billion t/year in 2020, with an average annual growth rate of about 7% [4]. Tailing production accounts for about 80% of total industrial waste, and the total reserves exceed 60 billion tons. At present, the main types of tailings at home and abroad are iron, copper, and gold, and the total stock of the three accounts for about 83% of the total tailings.

Located between 26°50′ and 36°53′ north latitude and between 78°25′ and 99°06′ east longitude, the Tibet Autonomous Region has an average altitude of over 4000 m. It is the main part of the Qinghai–Tibet Plateau, known as the “roof of the world”, with a vast territory. More than 3 million permanent residents are distributed on land that is 1.2 million square kilometers in size [5], and it is one of five ethnic autonomous regions in China. Tibet has special geology and is rich in natural resources, and its unique resource advantages and resource endowments give Tibet excellent metallogenic conditions [6]. With the acceleration of urbanization, the arrival of the era of industrialization, and the rich reserves of mineral resources, mineral resource mining, and tourism have become the mainstay of economic development in Tibet [7]. The exploitation of mineral resources not only drives economic development but also causes a series of environmental problems. For example, Wu et al. [8] found that the accumulation of tailings (slag) in southwest Guizhou seriously exceeded the standard of As, Cd, and Hg in the surrounding surface soil. In addition to the impact on the surrounding soil, mining will have an impact on water, the atmosphere, geology, etc. The tailing wastewater generated by mining will lead to the balance between groundwater, surface water, and precipitation being disrupted, and mining can easily result in soil erosion, mining area settlement, etc. In the process of quarry operations, a lot of soot and exhaust gas will be produced, thus causing serious pollution to the atmosphere [9,10,11]. Ruiz-Ortiz et al. [12] found that the surrounding environment of lead–zinc tailing ponds increased gradually with increasing distance from the tailing pond. Many studies have found that the accumulation of tailings will also affect the surrounding environment, and the development of tailing resource utilization has become the main way of solving the problem of large tailing reserves.

A total of 19 mineral deposits have been discovered in Lhasa, Tibet. Copper, lead, and zinc mines account for 78.95% of the total mines around Lhasa [13]. In order to deal with the slag generated by mining, there are currently seven copper, lead, and zinc tailing ponds in use and two copper tailing ponds under construction around Lhasa [14]. Tibet’s special geology and landform have resulted in the formation of many mineral resources (molybdenum ore, chromite, copper ore, lead, and zinc ore). Therefore, mining has become the mainstay of Tibet’s economic development. In addition to driving economic development, mining also causes a series of environmental problems. Under the prospect of simultaneous economic and ecological protection, tailing ponds have played a significant role in the treatment of tailings and slag, but long-term storage will also have an impact on the surrounding environment, especially the heavy metals in tailings.

Therefore, by investigating the background information of the typical tailing ponds around Lhasa, this study focused on the specific activity of radium-226, thorium-232, potassium-40, and other radioactive indicators in 17 mining areas, analyzed the relationship between radionuclides and radioactivity levels in each mining area, and evaluated the gamma radiation level exposure to miners and their surrounding residents.

## 2. Survey and Research Methods

### 2.1. Sample Collection and Processing

In this study, 17 typical metal tailing mining areas in Lhasa, Tibet, are considered, and the sampling points are calibrated by using directional Angle distribution points combined with GPS coordinate information. The 17 sampling points in the tailing pond are numbered 1–17. The specific distribution of sampling points is shown in Figure 1, and the specific mining areas of the sampling points are shown in Table 1.

Samples were dried in an oven at 200 °C for 72 h until a constant mass (103 g) was obtained to ensure that moisture was completely removed from the sample to avoid clamping the sample particles during crushing. The sample was then pulverized into a powder using a mortar and pestle to increase the surface area. A 2 mm sieve was used to obtain a uniform particle size. The sieved samples were weighed with an electronic balance to determine their dry mass. The samples were then sealed in standard 500 mL beakers and stored for 30 days to achieve long-term equilibrium between radon and its progeny.

### 2.2. Radioactivity Detection

#### 2.2.1. Calibration of Detector

Gamma rays and their energy were detected using a sodium iodide detector (GDM 20 series). First, the gamma-ray detector was calibrated. Europium (Eu-152) radioisotope was used as the reference material to calibrate the energy and efficiency of the detector. Eu-152 was used in the calibration process because it emits many gamma rays of known intensity and energy. The spectrum of the Eu-152 solution was obtained by placing it on the detector for 6000 s. AutoDas software 2020 was used to analyze the peak of known energy, and the channel number scale was converted into an energy scale. The resolution of the detector was determined by energy and efficiency calibration. To obtain the background radiation, an empty 500 mL beaker was placed on the detector to produce a spectrum with a period of 6000 s. The background radiation spectrum was then subtracted from the gamma-ray spectrum of the sample to obtain a net count [15,16,17]. In addition, the setup was coupled with a computer-based multichannel analyzer (MCA), which will be used for data acquisition and analysis of gamma spectra.

#### 2.2.2. Identification of Radionuclides

Each sample was placed on a gamma-ray detector surrounded by a 10-cm-thick lead cylinder with a removable top to reduce background radiation. A minimum of 6000 s were counted in order to obtain a clear spectrum. Using AutoDas software, the gamma ray energy and intensity of each sample spectrum were obtained. Each energy peak observed in the gamma spectrum was compared with the standard energy peak of various radionuclides found in nature. The radionuclides produced in the energy peaks observed in the spectra of each sample were identified [17,18,19].

#### 2.2.3. Determination of Radon Concentration

Radon concentration was measured in metal tailing samples from 17 areas in Lhasa, Tibet, by RAD on an electronic detector (RAD7). It was measured in the air. The detector uses a solid-state alpha detector to convert alpha radiation into an electrical signal and has the ability to electrically determine the energy of each alpha particle. Radon was quantified from 218 Po and 214 Po peaks with α energies of 6.00 MeV and 7.69 MeV, respectively. Before beginning to measure radon, it was ensured that a desiccant and inlet filter was placed between the sample and RAD7. RAD7 is set to “sniff” mode because of its rapid response to concentration changes and its ability to recover quickly after high concentrations. Before the counting procedure, the pump was run for 5 min to flush the measuring chamber [20]. The actual measurement of radon concentration was performed over a full cycle, lasting 5 min, for a total of 48 cycles. At the end of the run, RAD7 displayed a bar chart showing the maximum, minimum, and average radon concentrations.

#### 2.2.4. Determination of Specific Activity

The activity level of each radionuclide determined in the sample was determined by subtracting the background radiation count from the total light peak area. The activity of ^226^Ra was determined to be 351.9 keV of 214 Bi and 609.3 keV of 214 Pb. The activity of 232 Th was measured using 238.6 keV of its decay product ^212^Pb and 583.2 keV of 208 Tl, while the concentration of ^40^K activity was measured in it’s own the gamma spectrum of 1460.8 keV [21,22,23,24].

The specific activity concentration levels (SAC, Bq/kg) of ^226^Ra, ^232^Th, and ^40^K in each sample were calculated using the following formula:(1)$$\mathrm{SAC}=\frac{N/T}{m\times \kappa \times \eta }\left(\mathrm{Bq}/\mathrm{kg}\right)$$
where N/T is the net activity rate, N is the net light peak area, m is the sample mass, k is the absolute γ emission probability (branching ratio), η is the detector efficiency, and T is the counting time for each sample. According to the standard radionuclide data sheet of the International Atomic Energy Agency [25], the branching ratio of each radionuclide was determined. By using this equation, the detector efficiency η was determined:(2)$$\eta =\frac{Y}{\mathrm{AT}}$$
where Y is the net peak area of the calibration spectrum, A is the activity of Eu-152 (reference radionuclide), and T is the counting time of the calibration spectrum.

#### 2.2.5. Determination of γ Dose Rate

The gamma radiation dose rate (D_O_ nGy/h) in air at 1 meter above ground level was calculated from the measured specific activity concentration using the following equation:(3)$$D_{O}=0.462A_{\mathrm{Ra}}+0.604A_{\mathrm{Th}}+0.0417A_{k}$$

Among them, 0.462, 0.604, and 0.0417 are the conversion factors of the absorbed dose rate of ^226^Ra, ^232^Th, and ^40^K, respectively, and A_Ra_, A_Th_, and A_K_ are their average specific activities, respectively [26,27,28,29,30].

The annual outdoor effective dose caused by ground gamma radiation can be calculated using the conversion coefficient between the absorbed dose in air and the effective dose in the human body denoted as 0.7 Sv G/y, and the outdoor occupancy coefficient, denoted as 0.2. The formula is as follows:(4)$$\mathrm{EO}=D_{o}\times 0.7\times 0.2\times T$$
where EO is the annual outdoor effective dose (mSv/y), D_O_ is the absorbed dose rate in air, and T is 8760 h [31,32,33,34]. 

### 2.3. Determination of Radiation Hazard Index

The radiation hazard index is determined to assess the level of exposure to γ-rays that can cause harm to the exposed person. Therefore, the external hazard index, internal hazard index, and γ index were determined in this study.

We assume that a dose of 370 Bq/kg of ^226^Ra, 259 Bq/kg of ^232^Th, and 1410 Bq/kg of ^40^K yields the same γ dose. When the human body is exposed to gamma rays, the tissues of the body absorb a certain amount of gamma rays. How much radiation is absorbed depends on how the body is exposed to radiation. The damage from radiation depends on many factors, including the amount of radiation absorbed, the duration of exposure, and the person’s age and sex.

External risk index and internal risk index are two radiation risk indexes used to assess the risk of γ radiation exposure, and their maximum values must be less than 1 (unit).

Due to the influence of external gamma radiation, the external hazard index (H_ex_) can be calculated using Equation (5) [35,36,37]:(5)$$H_{\mathrm{ex}}=\frac{A_{\mathrm{Ra}}}{370}+\frac{A_{\mathrm{Th}}}{259}+\frac{A_{K}}{4810}$$

The internal hazard index (H_in_) caused by gamma radiation from internally deposited radionuclides is calculated using Equation (6) [38,39,40]:(6)$$H_{\mathrm{in}}=\frac{A_{\mathrm{Ra}}}{185}+\frac{A_{\mathrm{Th}}}{259}+\frac{A_{K}}{4810}$$

The gamma index is another hazard index used to screen materials containing ^226^Ra, ^232^Th, and ^40^K that could become hazardous when used as construction materials. The γ index (Iγ) was calculated using Formula (7):(7)$$I_{\gamma }=\frac{A_{\mathrm{Ra}}}{300}+\frac{A_{\mathrm{Th}}}{200}+\frac{A_{K}}{3000}$$

For commonly used materials such as concrete and sand, the standard dose rate for I_γ_ < 0.5 is 0.3 mSv/y. If I_γ_ > 6, the material cannot be used [41,42,43].

## 3. Results and Discussion

### 3.1. Total αβχγ Radiation and Radon Concentration

The total αβχγ radiation and radon concentration of metal tailings from the 17 mining areas in Lhasa, Tibet, is shown in Figure 2 and Table 2.

The results show that the radiation doses of the 17 tailing dumps ranged from 0.08 μSv/h to 0.26 μSv/h, and the largest is located in the tailing pond of the concentrator of Xizang Jinlianda Mining Co., Ltd. (Lhasa, China) In this survey, the occupational exposure dose limit for external environmental exposure to ionizing radiation is 20 mSv/a, and for the public, it is 1 mSv/a(GB18871-2002). The annual working hours recommended by ICRP (2000 h/a) for occupational exposure and 8760 h for public exposure were adopted, and the outdoor residence factor was considered to be 0.2. The calculated public and occupational radiation doses were 0.45 mSv/a and 0.52 mSv/a, respectively. The calculated results showed that the effective doses for public and occupational exposure in all mining areas did not exceed the annual dose limits (1 mSv/a and 20 mSv/a) stipulated in the basic standards of ionizing radiation protection and radiation source safety (GB 18871-2002). The research results show that the concentration range of radon in the air of the tailing yards in the 17 mining areas is 10.8 Bq/m^3^–29.6 Bq/m^3^. According to the national standard GB50325-2020 “Indoor Environmental Pollution Control Standard for Civil Construction Engineering”, the upper limit of indoor radon in civil buildings is 100 Bq/m3. The results show that radon concentration in the air of mining areas is far below the standard value.

In conclusion, the radon concentration, total (αβχγ) radiation, and total (χγ) radiation in the air of the 17 tailing dumps in the surrounding areas of Lhasa do not exceed the national radiation-related standards, and the environmental hazard risk is low.

### 3.2. Specific Activity Concentration

Various spectra were generated from the spectral analysis of the samples and were used to identify the different radionuclides in each sample. Radionuclides in the radium (^226^Ra), thorium (^232^Th), and potassium (^40^K) series were identified from spectral and activity calculations. The specific activity concentrations of ^226^Ra, ^232^Th, and ^40^K in the 17 mining samples were obtained using Equations (1) and (2), and the test results are shown in Table 3. The specific activity concentration distributions of ^226^Ra, ^232^Th, and ^40^K in the 17 mining samples are shown in Figure 3. The results show that the specific activity concentration of ^226^Ra ranged from 8.91 Bq/kg to 94.61 Bq/kg, and the specific activity concentration of ^232^Th ranged from 2.90 Bq/kg to 89.62 Bq/kg. The specific activity concentration of ^40^K ranged from less than MDA to 762.89 Bq/kg. In this study, the specific activity concentration values of ^226^Ra, ^232^Th, and ^40^K around the world are shown in Table 4. The average specific activity values of ^232^Th and ^40^K in Lhasa, Tibet, are lower than the global average and the Chinese average for global tailing samples, while ^226^Ra was slightly higher than the global average and the Chinese average, and the average specific activity concentration of ^40^K is the highest. In addition, the average specific activity concentration of ^226^Ra is lower than that of ^232^Th, and the average specific activity concentration of ^40^K is almost an order of magnitude higher than that of ^232^Th.

### 3.3. γ Radiation Dose Level

The average outdoor absorbed dose rate (DO) and the average annual outdoor effective dose rate (EO) at 1 m above the ground were calculated using Equations (3) and (4), respectively. The calculation results are shown in Table 5.

According to the calculation results in Table 5, the average DO of the 17 mining areas in Lhasa, Tibet, is 39.82 nGy/h, which is lower than the global average of 53.46 nGy/h and the Chinese average of 77.50 nGy/h. The average EO of the 17 mining areas is 0.05 mSv/y, which is lower than the global average of 0.07 mSv/y and the Chinese average of 0.10 mSv/y. It can be concluded that the dose level of γ radiation in the 17 mining areas is safe. The average ELCR of the 17 mining areas is 0.17, which is lower than the global average of 0.23 and the Chinese average of 0.33, so there will be little increase in γ-radiation-induced cancer cases and other health hazards among exposed miners and people living around the mines.

### 3.4. Radiation Hazard Index

Because soil can be used as a building material, exposure to outdoor gamma radiation and exposure to building materials containing ^226^Ra, ^232^Th, and ^40^K can be dangerous to residents of such houses and buildings. Therefore, the radium equivalent activity (Ra_eq_), external hazard index (H_ex_), internal hazard index (H_in_), and γ index (I_γ_) were measured to assess the level of gamma radiation hazard regarding natural radionuclides in mine soil tailings when used as building materials. Equation (5) was used to calculate the radium equivalent activity of the soil tailings. The external risk index, internal risk index, and γ risk index were determined using Equations (6), (7), and (8), respectively. The results are shown in Table 6.

It can be seen from Table 6 that the average external risk index of the 17 mining areas was 0.24, and the average internal risk index was 0.34, which are both lower than the maximum allowable limit of the unit. This means that the metal tailings from all 17 mining areas are within the limit for γ radiation such that the hazards from external γ radiation exposure and the internal deposits of radionuclides (mainly from radon exposure) are not significant. The average value of the γ index of metal tailings in the 17 mining areas is 0.31, which is lower than 1. Therefore, these materials can be used as major building materials in batches.

## 4. Conclusions

In this paper, the natural radioactivity levels of metal tailings in 17 typical metal tailing areas in Lhasa, Tibet, were studied, and the following conclusions were drawn:(1)The radiation doses ranged from 0.08 μSv/h to 0.26 μSv/h, and the radon concentration ranged from 10.8 Bq/m3 to 29.6 Bq/m3, none of which exceeded the national radiation-related standards, indicating a low environmental hazard risk;(2)The specific activity concentration of ^226^Ra ranged from 8.91 Bq/kg to 94.61 Bq/kg, and the specific activity concentration of ^232^Th ranged from 2.90 Bq/kg to 89.62 Bq/kg. The specific activity concentration of ^40^K ranged from less than MDA to 762.89 Bq/kg. The average specific activities of ^232^Th and ^40^K were less than the global average and the Chinese average for global tailing samples, while ^226^Ra was slightly higher than the global average and the Chinese average;(3)The average absorbed dose rate (DO) of the 17 mining areas was 39.82 nGy/h, and the average annual effective dose rate (EO) was 0.057 mSv/y, which barely increases the cancer cases and other health hazards induced by γ radiation;(4)The average external risk index of the 17 mining areas was 0.24, the average internal risk index was 0.34, and the average γ index was 0.31, which were all less than the maximum permissible limit. This means that the metal tailings from all 17 mining areas were within the limit for γ radiation and, therefore, can be used in bulk as major building materials without posing a significant radiation threat to the residents of the study area.

## Figures and Tables

**Figure 1 ijerph-20-04525-f001:**
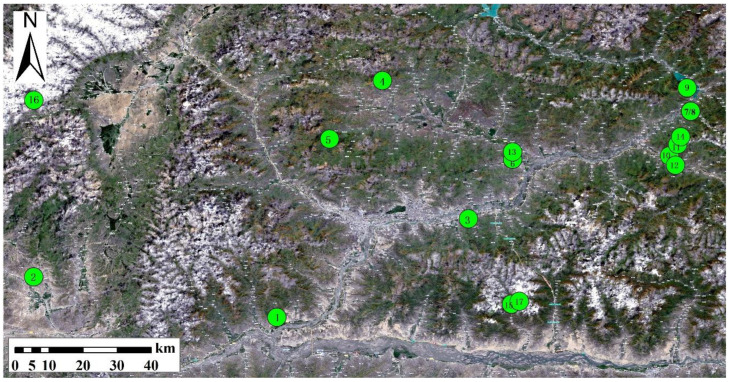
Sampling point map of metal tailings.

**Figure 2 ijerph-20-04525-f002:**
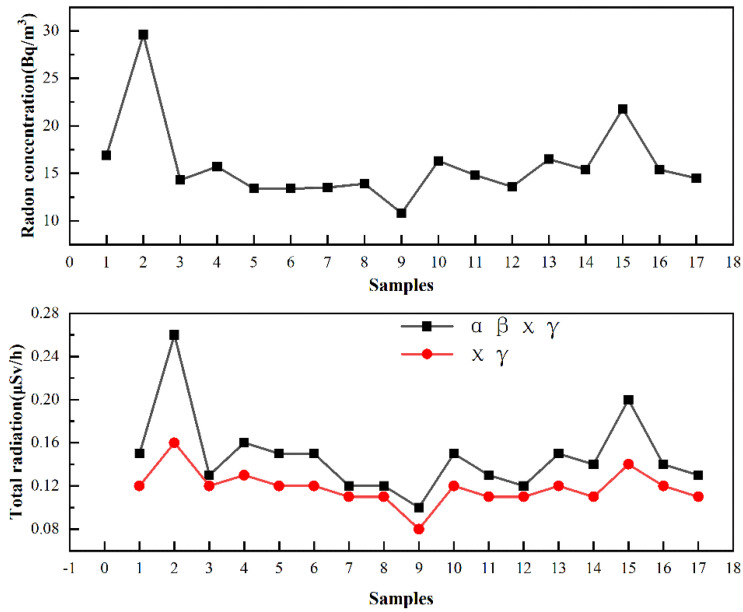
Radioactivity detection of tailing yard in the 17 mining areas.

**Figure 3 ijerph-20-04525-f003:**
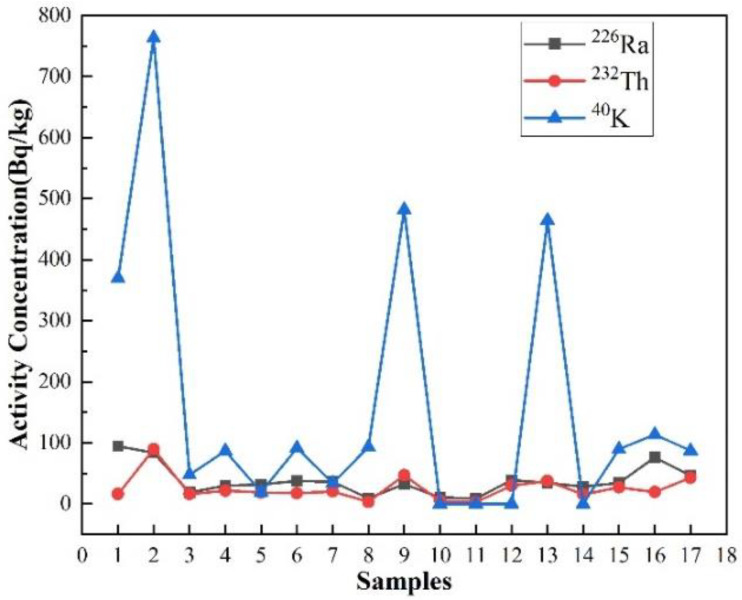
Specific activities of ^226^Ra, ^232^Th, and ^40^K in 17 mining areas.

**Table 1 ijerph-20-04525-t001:** Specific mining situation of metal tailing sampling sites in Lhasa, Tibet.

Sampling Point	Site Selection of Tailing Pond	Temperature	Humidity	Altitude	Longitude	Latitude
1	Qushui county Jigong village molybdenum ore dressing plant tailing pond	17 °C	55%	3668 m	90.774523	29.38994
2	Tailing pond of concentrator of Xizang Jinlianda Mining Co., Ltd.	23 °C	31%	4081 m	90.132549	29.496971
3	Tailing pond of Dazipuxiong copper concentrator	12 °C	73%	3702 m	91.280719	29.64992
4	Jialapu iron mine tailing pond of Xizang Yexin Mining Co., Ltd.	10 °C	91%	4400 m	91.053419	30.014964
5	Tailing pond of Lunlang lead–zinc mine, Kazi township, Xiruide Mining Company	10 °C	90%	4330 m	90.913583	29.860757
6	Tailing pond of dressing plant of Caisheng Mining Co., Ltd., Jiangxia Township, Linzhou County	17 °C	59%	3772 m	91.397533	29.806781
7	Tailing pond of Zhangda Phase II concentrator, Nyima Jiangre Township, Zhongkai Mining	11 °C	77%	3905 m	91.867207	29.933872
8	Nimajiang Rexiang Phase I tailing Pond, Zhongkai Mining	11 °C	77%	3905 m	91.867329	29.93384
9	Tangjia lead–zinc concentrator tailing pond	16 °C	54%	3891 m	91.858273	29.99531
10	Tailing pond of Zhaxigang Township Copper, Lead and Zinc Ore concentrator of Yuanze Mining Co., Ltd.	22 °C	32%	3915 m	91.811313	29.816292
11	Mozhugong Kawanyang Industry and Trade Company lead–zinc tailing pond	21 °C	32%	3919 m	91.833212	29.846291
12	Hesheng Mining tailing pond	21 °C	31%	3917 m	91.827396	29.7916281
13	Jingcheng Mining Tailing Pond (acquired by Wanyang)	23 °C	30%	3913 m	91.397321	29.826191
14	Xizang Mozhu Baoyuan Lead–zinc concentrator tailing pond	23 °C	30%	3911 m	91.841212	29.866212
15	Tailing pond of Qulong Copper polymetallic concentrator in Mozhugongka County, Xizang Julong Copper Industry Co., Ltd.	12 °C	79%	4010 m	91.394323	29.424967
16	Xizang Duilong Dongwei Industrial Co., Ltd. (Tailing Pond, Bangcun, Deqing Town)	17 °C	42%	4030 m	90.132743	29.962931
17	Phase II of Jiama Copper polymetallic concentrator of Huatailong Mining Company	16 °C	54%	4330 m	29.432125	91.415323

**Table 2 ijerph-20-04525-t002:** Measurement of radioactive absorbed dose rate of tailing yard in 17 mining areas.

Samples	1	2	3	4	5	6	7	8	9	10	11	12	13	14	15	16	17
Total αβχγ radiation (μSv/h)	0.15	0.26	0.13	0.16	0.15	0.15	0.12	0.12	0.1	0.15	0.13	0.12	0.15	0.14	0.2	0.14	0.13
Total χγ radiation (μSv/h)	0.12	0.16	0.12	0.13	0.12	0.12	0.11	0.11	0.08	0.12	0.11	0.11	0.12	0.11	0.14	0.12	0.11
Radon concentration (Bq/m^3^)	16.9	29.6	14.3	15.7	13.4	13.4	13.5	13.9	10.8	16.3	14.8	13.6	16.5	15.4	21.8	15.4	14.5

**Table 3 ijerph-20-04525-t003:** Specific activity concentrations of ^226^Ra, ^232^Th, and ^40^K in the 17 mining samples (Bq/kg).

Samples	1	2	3	4	5	6	7	8	9	10	11	12	13	14	15	16	17	Average
^226^Ra	94.62	83.88	19.03	30.18	31.70	37.67	36.13	9.09	32.24	10.97	8.91	39.06	34.26	28.45	34.42	76.26	46.27	38.42
^232^Th	16.09	89.62	16.07	21.58	18.81	17.55	20.55	3.42	46.97	5.12	2.90	30.11	37.72	15.74	27.26	19.44	42.76	25.40
^40^K	369.66	762.89	47.88	86.95	19.44	91.64	33.66	93.71	482.24	-	-	-	464.49	0.00	89.65	113.80	87.43	161.38

**Table 4 ijerph-20-04525-t004:** Comparison of natural radioactivity of metal tailings in Tibet and other parts of the world [44].

Location	India [45]	Nigeria [46]	Pakistan [47]	Jordan [48]	Turkey [49]	Greece [50]	Japan [51]	USA [51]	Global Average [51]	China Average
^226^Ra	27.9	33.6	49.0	42.5	33	27	33	40	35.75	25.1
^232^Th	72.8	20.1	62.4	26.7	32	36	28	35	39.13	67.1
^40^K	287	207	671	291	255	496	310	37	319.25	608.6

**Table 5 ijerph-20-04525-t005:** Mean outdoor absorbed dose rate DO (nGy/h) and mean annual outdoor effective dose rate EO (mSv/y).

Samples	1	2	3	4	5	6	7	8	9	10	11	12	13	14	15	16	17	Average	Global Average	China Average
DO	68.84	124.70	20.49	30.60	26.82	31.83	30.51	10.18	63.37	8.16	5.87	36.24	57.98	22.65	36.10	51.72	50.85	39.82	53.46	77.50
EO	0.08	0.15	0.03	0.04	0.03	0.04	0.04	0.01	0.08	0.01	0.01	0.04	0.07	0.03	0.04	0.06	0.06	0.05	0.07	0.10

**Table 6 ijerph-20-04525-t006:** H_ex_, H_in_, and I_γ_ of ^226^Ra, ^232^Th, and ^40^K in the 17 mining samples.

Samples	1	2	3	4	5	6	7	8	9	10	11	12	13	14	15	16	17	Average	Global Average	China Average
H_ex_	0.39	0.73	0.12	0.18	0.16	0.19	0.18	0.06	0.37	0.05	0.04	0.22	0.33	0.14	0.22	0.30	0.31	0.24	0.31	0.45
H_in_	0.65	0.96	0.17	0.26	0.25	0.29	0.28	0.08	0.46	0.08	0.06	0.33	0.43	0.21	0.31	0.51	0.43	0.34	0.41	0.52
I_γ_	0.52	0.98	0.16	0.24	0.21	0.24	0.23	0.08	0.50	0.06	0.04	0.28	0.46	0.17	0.28	0.39	0.40	0.31	0.42	0.62

## Data Availability

The data presented in this study are available from the corresponding author upon request.

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
