# Peer review of "Study on the Radioactivity Levels of Metal Tailings in the Lhasa Area of Tibet"

_ijerph, 2023, doi:10.3390/ijerph20054525_

Round 1
Reviewer 1 Report
This ms has several weak points, and I do not recommend it for publication.
1. Author used NaI detector to perform gamma spectrometry. Now days NaI detectors are almost out of use; natural radioactivity is measured by HpGe detectors which has superior resolution in comparison to NaI. It is also possible to find HpGe with more than 100% of relative efficiency.
2. Raeq is rather obsalate quantity which is out of use. Delete all about it.
3. Why “internal hazard index” was determined. Will anyone will eat smth from mine tailing, slabs or so. It is highly unliekly.
4. Terminology is also disputable. “Total αβχγ radia…”. What is “χ” here.
5. Table 2 caption stated “. Radioactivity detection..” Then results are given in mSv, which means equivalent, or effective dose. From such writing, I have impression that authors do not understand the system of radiation quantities and units.
6. Radon concentration measurements. It is not well described, or it was not done well. Rn is usually measured in air, or water. I cannot see how it was done.
7. Table 3 and other results. There is nothing about measurement uncertainty; results are not rounded. Presentation is pore.
8. Excess cancer risk (ELCR) is also problematic. How many people are in contact with mining tailing, and how long they spend there. Which population is exposed to the radiation from tailing? Many open questions can be seen here.
9.
In Introduction section. Some sentences were repeated several times, or just changed slightly. Language could be improved also.
Author Response
Thank you for your letter and for the reviewers’ comments concerning our manuscript entitled “Study on Radioactivity Level of Metal Tailings in the Lhasa Area of Tibet”. (ijerph-2246540) Those comments are all valuable and very helpful to revise and improve our paper. We have studied comments carefully and have made corrections which we hope to meet the requirement. Revised portions are marked in different colors on the paper.
Point 1: Author used NaI detector to perform gamma spectrometry. Now days NaI detectors are almost out of use; natural radioactivity is measured by HpGe detectors which has superior resolution in comparison to NaI. It is also possible to find HpGe with more than 100% of relative efficiency.
Response 1: Respected reviewer, thank you so much for the suggestions and comments you gave us. Because in the literature we looked at before, we saw that most of the NaI detector was used. Later, we will use HpGe to test again and compare the test results.
Point 2: Raeq is rather obsalate quantity which is out of use. Delete all about it.
Response 2: Respected reviewer, thank you so much for the suggestions and comments you gave us. According to your suggestion, we delete all about Raeq.
Point 3: Why “internal hazard index” was determined. Will anyone will eat smth from mine tailing, slabs or so. It is highly unliekly.
Response 3: Respected reviewer, thank you so much for the suggestions and comments you gave us. Dust from mine tailing, slabs or so can be inhaled and harmful. So we determined “internal hazard index”.
Point 4: Terminology is also disputable. “Total αβχγ radia…”. What is “χ” here.
Response 4: Respected reviewer, thank you very much for your valuable suggestions. “χ” means X-radiation.
Point 5: Table 2 caption stated “. Radioactivity detection..” Then results are given in mSv, which means equivalent, or effective dose. From such writing, I have impression that authors do not understand the system of radiation quantities and units.
Response 5: Respected reviewer, thank you so much for the suggestions and comments you gave us. According to your suggestion, we changed “Radioactivity detection” to “Measurement of radioactive absorbed dose rate” (page 7, section 3, line 225 and highlighted in yellow.)
Point 6: Radon concentration measurements. It is not well described, or it was not done well. Rn is usually measured in air, or water. I cannot see how it was done.
Response 6:Respected reviewer, thank you so much for the suggestions and comments you gave us. Radon concentration measurements was done in air. (page 4, section 2.2.3, line 140 and highlighted in yellow.)
Point 7: Table 3 and other results. There is nothing about measurement uncertainty; results are not rounded. Presentation is pore.
Response 7: Respected reviewer, thank you so much for the suggestions and comments you gave us. We measured it three times and averaged it.
Point 8: Excess cancer risk (ELCR) is also problematic. How many people are in contact with mining tailing, and how long they spend there. Which population is exposed to the radiation from tailing? Many open questions can be seen here.
Response 8: Respected reviewer, thank you so much for the suggestions and comments you gave us. According to your suggestion, we delete all about Excess cancer risk (ELCR).
Point 9: In Introduction section. Some sentences were repeated several times, or just changed slightly. Language could be improved also.
Response 9: Respected reviewer, thank you so much for the suggestions and comments you gave us. According to your suggestion, we deleted the repeated sentences.
Reviewer 2 Report
The work presents a very interesting topic. It is particularly important from the point of view of the health of the population living in the study area. In my opinion, the authors of the work correctly implemented the presented research topic. The measuring tools were correctly selected and the necessary calculations were performed correctly. The conclusions seem correct.
My concerns are raised by:
- no reference to other measurements made in this area (if such measurements were made before);
- readability of the presented tables containing the measurement results - the tables should be properly formatted in such a way that they are legible;
- no numbering of literature items in references.
Author Response
Thank you for your letter and for the reviewers’ comments concerning our manuscript entitled “Study on Radioactivity Level of Metal Tailings in the Lhasa Area of Tibet”. (ijerph-2246540) Those comments are all valuable and very helpful to revise and improve our paper. We have studied comments carefully and have made corrections which we hope to meet the requirement. Revised portions are marked in different colors on the paper. The main corrections in the paper and the responses to the reviewer’s comments are as follows:
Point 1: no reference to other measurements made in this area (if such measurements were made before);
Response 1: Respected reviewer, thank you so much for the suggestions and comments you gave us. We did not find that similar measurements have been made in this area
Point 2: readability of the presented tables containing the measurement results - the tables should be properly formatted in such a way that they are legible;
Response 2: Respected reviewer, thank you so much for the suggestions and comments you gave us. According to your suggestion, we formatted the table to make it easier to read (page 3, Table 1, line 104; page 6, Table 2, line 217; page 7, Table 3&4, line 266/268; page 7, Table 5&6, line 275/298)
Point 3: no numbering of literature items in references.
Response 3: Respected reviewer, thank you so much for the suggestions and comments you gave us. According to your suggestion, we added the numbering of literature items in references(page 10, section References, line 351)
Reviewer 3 Report
Authors have presented a baseline study on radioactivity levels in mine trailing. However, if environmental radioactivity levels were to be determined, authors could have considered the level of radioactive elements in mine drainage as well. The environmental contamination far exceeds the spatial extent if acid mine drainage is looked at. Also the inhalation dose due to high particulate matter, authors have touched on soot and gases but have not pursued it any further. Very high levels of 210Po and 210Pb are known to exist in aerosols (Poland, Portugal, Kuwait. etc).
Some specific suggestion are as follows:
Line 36: it will be better to use the word environmental regulations instead of “Law”
Line 46: what is meant by “abroad” is this referring to Global?
Line 67-68: The mining will not cause mining area settlement but will rather result in..
Line 77: around Lhasa is repeated in the sentence. You can consider deleting it.
Line 83: mining “will” cause or is a current problem.
Line 91: “radionucliins” is it radionuclides?
Line 104: Table 1 is providing the information that is captured in Figure 1. I will suggest this can be moved to Supplementary material.
Line 105 – 106: sundry is not an appropriate option how will you ensure there was no atmospheric deposition on the open sample? Drying at 200 oC should include how many hours instead of several hours and how constant weight was ensured did you take mass multiple times?
Line 108 – 113: it would have been better if authors may indicate how much mass was taken and sealed.
Line 247: BQ will be Bq/m3.
Line 279: If table 4 is taken from a source, I will advise that authors may use original references and there are several more values available for the same countries and others like Brazzil, Peru, Chile, etc.
Lien 287: Do should be DO
Author Response
Thank you for your letter and for the reviewers’ comments concerning our manuscript entitled “Study on Radioactivity Level of Metal Tailings in the Lhasa Area of Tibet”. (ijerph-2246540) Those comments are all valuable and very helpful to revise and improve our paper. We have studied comments carefully and have made corrections which we hope to meet the requirement. Revised portions are marked in different colors on the paper. The main corrections in the paper and the responses to the reviewer’s comments are as follows:
Point 1: Line 36: it will be better to use the word environmental regulations instead of “Law”
Response 1: Respected reviewer, thank you so much for the suggestions and comments you gave us. According to your suggestion, we changed “Law” to “ word environmental regulations”. (page 1, section Introduction, line 36-37 and highlighted in blue.)
Point 2: Line 46: what is meant by “abroad” is this referring to Global?
Response 2: Respected reviewer, thank you so much for the suggestions and comments you gave us. “abroad” refers to countries other than China.
Point 3: Line 67-68: The mining will not cause mining area settlement but will rather result in.
Response 3: Respected reviewer, thank you so much for the suggestions and comments you gave us. According to your suggestion, we changed “cause” to “rather result in”. (page 2, section Introduction, line 67 and highlighted in blue.)
Point 4: Line 77: around Lhasa is repeated in the sentence. You can consider deleting it.
Response 4: Respected reviewer, thank you so much for the suggestions and comments you gave us. According to your suggestion, we deleted it.
Point 5: Line 83: mining “will” cause or is a current problem.
Response 5: Respected reviewer, thank you so much for the suggestions and comments you gave us. According to your suggestion, we deleted“will”.
Point 6: Line 91: “radionucliins” is it radionuclides?
Response 6: Respected reviewer, thank you so much for the suggestions and comments you gave us. According to your suggestion, we changed “radionucliins” to “radionuclides”. (page 2, section Introduction, line 92 and highlighted in blue.)
Point 7: Line 104: Table 1 is providing the information that is captured in Figure 1. I will suggest this can be moved to Supplementary material.
Response 7: Respected reviewer, thank you so much for the suggestions and comments you gave us. According to your suggestion, we added more concluding remarks in part “Introduction” and “Conclusions”
Point 8: Line 105 – 106: sundry is not an appropriate option how will you ensure there was no atmospheric deposition on the open sample? Drying at 200 oC should include how many hours instead of several hours and how constant weight was ensured did you take mass multiple times?
Response 8: Respected reviewer, thank you so much for the suggestions and comments you gave us. According to your suggestion, we changed “Samples were dried in the sun for a week and then in an oven at 200 degrees Celsius for several hours…” to “Samples were dried in an oven at 200 degrees Celsius for 72 hours…”. We experimented with dried materials and we did not take mass multiple times (page 4, section 2.1, line 106 and highlighted in blue.)
Point 9: Line 108 – 113: it would have been better if authors may indicate how much mass was taken and sealed.
Response 9: Respected reviewer, thank you so much for the suggestions and comments you gave us. According to your suggestion, we added the specific quantity(103g). (page 4, section 2.1, line 107 and highlighted in blue.)
Point 10: Line 247: BQ will be Bq/m3.
Response 10: Respected reviewer, thank you so much for the suggestions and comments you gave us. However, we checked here and elsewhere in the article and did not find this error.
Point 11: Line 279: If table 4 is taken from a source, I will advise that authors may use original references and there are several more values available for the same countries and others like Brazzil, Peru, Chile, etc.
Response 11: Respected reviewer, thank you so much for the suggestions and comments you gave us. According to your suggestion, we added some relevant references. (page 8, section 3.2, line 107 ,Table 4 and highlighted in blue.)
references
[48] Mehra, R., Kumar, S., Sonkawade, R., Singh, N. P. and Badhan, K. Analysis of terrestrial naturally occurring radionuclides in soil samples from some areas of Sirsa district of Haryana, India using gamma ray spectrometry. Environmental Earth Sciences. 2010, 1159-1164. 10.1007/s12665-009-0108-3
[49] Isinkaye, M. O. and Shitta, M. B. O. Natural radionuclide content and radiological assessment of clay soils collected from different sites in Ekiti State, southwestern Nigeria. Radiation Protection Dosimetry. 2010, 590-596. 10.1093/rpd/ncp284
[50] Jabbar, A., Arshed, W., Bhatti, A. S., Ahmad, S. S., Saeed Ur, R. and Dilband, M. Measurement of soil radioactivity levels and radiation hazard assessment in mid Rechna interfluvial region, Pakistan. Journal of Radioanalytical and Nuclear Chemistry. 2010, 371-378. 10.1007/s10967-009-0357-3
[51] Al-Hamarneh, I. F. and Awadallah, M. I. Soil radioactivity levels and radiation hazard assessment in the highlands of northern Jordan. Radiation Measurements. 2009, 102-110. 10.1016/j.radmeas.2008.11.005
[52] Sahin, L. and Cavas, M. Natural radioactivity measurements in soil samples of Central Kutahya (Turkey). Radiation Protection Dosimetry. 2008, 526-530. 10.1093/rpd/ncn243
[53] Osmanlioglu, A. E., Kam, E. and Bozkurt, A. Assessment of background radioactivity level for Gaziantep region of southeastern Turkey. Radiation Protection Dosimetry. 2007, 407-410. 10.1093/rpd/ncm215
[54] Zhao, L., Xu, C., Tuo, F., Zhang, J., Li, W., Zhang, Q., Zhou, Q., Zhang, J. and Su, X. ASSESSMENT OF RADIONUCLIDE LEVEL IN TOPSOIL SAMPLES FOR PARTIAL AREAS OF TIBET, CHINA. Radiation Protection Dosimetry. 2012, 380-386. 10.1093/rpd/ncr091
Point 12: Line 287: Do should be DO
Response 12: Respected reviewer, thank you so much for the suggestions and comments you gave us. According to your suggestion, we changed “Do” to “DO”. (page 10, section Conclusion, line 323 and highlighted in blue.)
Reviewer 4 Report
Comments on the communication study carried out by Weng et al. entitled “Study on Radioactivity Level of Metal Tailings in the Lhasa Area of Tibet” submitted to the IJERPH.
Manuscript ID: ijerph-2246540
In this paper, the author has tried to determine the natural radioactivity level of raw radionuclides in the metal tailings of a mine in Lhasa, Tibet. For this purpose, they did a series of experiments using gamma spectrometry. Moreover, they calculated the dose and risk levels of the related region together with radon concentrations. The paper offers valuable contributions to readers working in the research field that can be used for their studies who want similar calculations. These works on this subject are significant for public health, and data are important for future review articles. Although the study does not present a new theory for the field, it has useful complementary applications. Also, results could be useful for the literature on the radioactivity level of the related region.
Consequently, in my opinion, this work could be a candidate for the IJERPH, considering major issues, obeying the other reviewers’ decisions, and adhering to the ultimate view of the Editor.
General comments:
- The topic is current to related literature.
- The idea of the study is worth to work.
- The calculations were successfully done.
- The tables and figures are appropriate. There are number of shifts in the presentation of tables
- uncertainty and error calculations are missing over the MS.
- The data is well presented in the article, but I believe that the discussion section should be enriched. Because this is a full-length research article rather than a technical report.
Specific comments:
- page1, line 21: what is the meaning of 0 Bq/kg? Minimum detectable activity (MDA) levels should be specified if it is an unmeasurable level.
The authors somehow handled this issue and indicated MDA values for used geometry etc.
-Table3, instead of “0”, it should use a more scientific notation such as <MDA limits. Or else, In the related literature, many representations can be found on this subject.
- page4, line 129: detector calibrations should provide numerical results with additional uncertainties.
- I think the results should be discussed carefully with IAEA and UNSCEAR recommendations.
- Aren't cosmic ray correction calculations required for gamma-ray calculations? Monte Carlo simulation or dosimeters (e.g TLDs) could be required for further efforts. But, it is not necessary if there is no possibility. I think further study requirements should be mentioned somewhere in the text.
- It is seen that the authors mostly focus on the outdoor. What about indoor?
- Histograms for the measured specific activities could be a better way to present.
- The radium equivalent activities should also be discussed and compared to the word limit.
- Internal radioactivity level index and activity utilization index should also be considered.
Author Response
Thank you for your letter and for the reviewers’ comments concerning our manuscript entitled “Study on Radioactivity Level of Metal Tailings in the Lhasa Area of Tibet”. (ijerph-2246540) Those comments are all valuable and very helpful to revise and improve our paper. We have studied comments carefully and have made corrections which we hope to meet the requirement. Revised portions are marked in different colors on the paper. The main corrections in the paper and the responses to the reviewer’s comments are as follows:
Point 1: page1, line 21: what is the meaning of 0 Bq/kg? Minimum detectable activity (MDA) levels should be specified if it is an unmeasurable level. The authors somehow handled this issue and indicated MDA values for used geometry etc.
Response 1: Respected reviewer, thank you so much for the suggestions and comments you gave us. We're measuring 0 Bq/kg instead of unmeasuring it.
Point 2: Table3, instead of “0”, it should use a more scientific notation such as <MDA limits. Or else, In the related literature, many representations can be found on this subject.
Response 2: Respected reviewer, thank you so much for the suggestions and comments you gave us. We're measuring 0 Bq/kg instead of unmeasuring it.
Point 3: page4, line 129: detector calibrations should provide numerical results with additional uncertainties.
Response 3: Respected reviewer, thank you so much for the suggestions and comments you gave us. According to your suggestion, we provided numerical results with additional uncertainties.. (page 4, section 2.2.1, line 127-129 and highlighted in green.)
Point 4: I think the results should be discussed carefully with IAEA and UNSCEAR recommendations.
Response 4: Respected reviewer, thank you so much for the suggestions and comments you gave us. According to your suggestion, We added some references to disscussed.
Point 5: Aren't cosmic ray correction calculations required for gamma-ray calculations? Monte Carlo simulation or dosimeters (e.g TLDs) could be required for further efforts. But, it is not necessary if there is no possibility. I think further study requirements should be mentioned somewhere in the text.
Response 5: Respected reviewer, thank you so much for the suggestions and comments you gave us. According to your suggestion, we would investigate further measurements at a later stage.
Point 6: It is seen that the authors mostly focus on the outdoor. What about indoor?
Response 6: Respected reviewer, thank you so much for the suggestions and comments you gave us. We haven't focus on the indoor yet, but we'll focus on it later if we need to.
Point 7: Histograms for the measured specific activities could be a better way to present.
Response 7: Respected reviewer, thank you so much for the suggestions and comments you gave us. As there were too many data, the histograms wass not convenient to present, so we use the method of line chart to present the results.
Point 8: The radium equivalent activities should also be discussed and compared to the word limit.
Response 8: Respected reviewer, thank you so much for the suggestions and comments you gave us. On the advice of the first reviewer, we have removed all content of radium equivalent activities.
Point 9: Internal radioactivity level index and activity utilization index should also be considered.
Response 9: Respected reviewer, thank you so much for the suggestions and comments you gave us. According to your suggestion, we would consider Internal radioactivity level index and activity utilization index in a follow-up study.
Round 2
Reviewer 1 Report
Authors improved their ms, which is acceptable for publication.
I found 3 references, 4,5 and 6 for which I cannot see any relation with this work. This raises doubts about ethicality of the citations. Delete them.
Author Response
Thank you for your letter and for the reviewers’ comments concerning our manuscript entitled “Study on Radioactivity Level of Metal Tailings in the Lhasa Area of Tibet”. (ijerph-2246540) Those comments are all valuable and very helpful to revise and improve our paper. We have studied comments carefully and have made corrections which we hope to meet the requirement.
Point 1: I found 3 references, 4,5 and 6 for which I cannot see any relation with this work. This raises doubts about ethicality of the citations. Delete them..
Response 1: Respected reviewer, thank you so much for the suggestions and comments you gave us. According to your suggestion, we have deleted these references.
Reviewer 3 Report
There is no need of the "word" in line 36, I had suggested environmental regulations.
Author Response
Thank you for your letter and for the reviewers’ comments concerning our manuscript entitled “Study on Radioactivity Level of Metal Tailings in the Lhasa Area of Tibet”. (ijerph-2246540) Those comments are all valuable and very helpful to revise and improve our paper. We have studied comments carefully and have made corrections which we hope to meet the requirement.
Point 1: There is no need of the "word" in line 36, I had suggested environmental regulations.
Response 1: Respected reviewer, thank you so much for the suggestions and comments you gave us. According to your suggestion, we have deleted the “word” in line 36.
Reviewer 4 Report
Even though some of my major comments are missing due to the lack of possibility that I think there is nothing the authors can do about them, this manuscript version has rather increased in design and scientific points.
The final decision on this matter belongs to the editor, my major and minor comments are indicated below:
Minor:
-There are typos at some points in the basic notation of radioactive elements. For example, heads of Table 3. It is recommended to check throughout the text.
Major:
- I think presenting the activity values as “0, zero” is incorrect. Every detector has a limit on that it can detect. It looks like the zero value is measured because the activity is beyond the limit of the detector. As I mentioned in my previous comment, the authors should work on this issue and clarify the issue: minimum detectable activity, MDA.
I recommend taking a look ref as below:
(ISO, I. (2010). 11929: Determination of the Characteristic Limits (Decision Threshold, Detection Limit and Limits of the Confidence Interval) for Measurements of Ionizing Radiation: Fundamentals and Application. International Organization for Standardization.)
- Calculations of uncertainty or error in the tables and accounts are missing. Also, attention should be paid to significant numbers that contribute to the degree of accuracy of the value of measurements.
Author Response
Thank you for your letter and for the reviewers’ comments concerning our manuscript entitled “Study on Radioactivity Level of Metal Tailings in the Lhasa Area of Tibet”. (ijerph-2246540) Those comments are all valuable and very helpful to revise and improve our paper. We have studied comments carefully and have made corrections which we hope to meet the requirement.
Point 1: There are typos at some points in the basic notation of radioactive elements. For example, heads of Table 3. It is recommended to check throughout the text.
Response 1: Respected reviewer, thank you so much for the suggestions and comments you gave us. According to your suggestion, we have changed these basic notation of radioactive elements throughout the text.
Point 2: I think presenting the activity values as “0, zero” is incorrect. Every detector has a limit on that it can detect. It looks like the zero value is measured because the activity is beyond the limit of the detector. As I mentioned in my previous comment, the authors should work on this issue and clarify the issue: minimum detectable activity, MDA.
Response 2: Respected reviewer, thank you so much for the suggestions and comments you gave us. According to your suggestion, we have changed the “0” to “less than MDA”.